# Determining Allometry and Carbon Sequestration Potential of Breadfruit (*Artocarpus altilis*) as a Climate-Smart Staple in Hawai'i

**Chad Livingston and Noa Kekuewa Lincoln \***

Department of Tropical Plant and Soil Sciences, College of Tropical Agriculture and Human Resources, University of Hawai'i at Mānoa, Honolulu, HI 96822, USA; clivings@hawaii.edu
\* Correspondence: nlincoln@hawaii.edu

**Abstract:** Breadfruit (*Artocarpus altilis*) is an underutilized Pacific tree crop that has been highlighted as having substantial potential to contribute to global food security and climate-smart agriculture, including adaptation to and mitigation of climate change. To explore the carbon sequestration potential of breadfruit production, we characterize tree volume, wood density, carbon density, foliar biomass, and growth rates of breadfruit in Hawai'i. Strong relationships to trunk or branch diameter were displayed for wood density ($r^2$ 0.81), carbon density ($r^2$ 0.87), and foliar biomass ($r^2$ 0.91), which were combined to generate an allometric prediction of tree volume ($r^2$ 0.98) based on tree diameter at breast height. Growth rates, as measured by diameter at breast height, were well predicted over time when trees were classified by habitat suitability. We extrapolate potential breadfruit growth and carbon sequestration in above-ground biomass to the landscape scale over time. This study shows that breadfruit is on the low end of broadleaf tropical trees in moist and wet environments, but in an orchard can be expected to sequester ~69.1 tons of carbon per hectare in its above-ground biomass over a 20-year period.

**Keywords:** allometric equation; climate-smart agriculture; carbon farming; agroforestry; Hawai'i; biomass; indigenous crops; neglected and underutilized species

## 1. Introduction

Globally, agricultural activities and associated deforestation account for an estimated 24% of anthropogenic greenhouse gas (GHG) emissions [1] and agricultural expansion to support globalized food industry supply chains, which drives deforestation on a global level [2]. A central question for humanity is how to feed a growing human population in a way that minimizes ecosystem impact or even plays a role in regenerating degraded functions of ecosystems, including the sequestration and storage of carbon.

The global situation is not entirely dissimilar to the challenges faced by the early inhabitants of the remote Pacific Islands. As human populations on these islands grew, managing landscapes to produce food and other plant resources in ways that did not result in total ecosystem collapse was a matter of survival. Among the diverse agroecological strategies employed, forest management and arboriculture were critical, often dominant, forms of food and resource production that maintained the integrity and function of the ecosystem [3,4]. Breadfruit (*Artocarpus altilis*)—a long-lived tree that produces large, starchy, carbohydrate-rich fruits—featured prominently in these forest management strategies. Originating in Papua New Guinea, breadfruit trees were dispersed via human migration throughout Polynesia and beyond as an essential crop in the Pacific cornucopia [5,6].

Breadfruit, and breadfruit agroforestry, remain vastly understudied despite significant international recognition of its potential roles in developing climate-smart agriculture and addressing global hunger [7,8]. As a long-lived tropical tree species, breadfruit has considerable potential to contribute to climate-smart agriculture in terms of mitigation,

adaptation, and resilience. Breadfruit not only has the potential to sequester carbon in its biomass, but its cultivation often accompanies farming practices such as reduced- or no-tillage and cover- or co-cropping that can further increase carbon storage and sequestration below ground. For human nutrition, breadfruit is a complex carbohydrate associated with a strong profile of vitamins, minerals, and amino acids [9,10]. From a socioeconomic perspective, tree crops support home food security and can reduce annual farm labor to support quality-of-life outcomes.

Because of its potential for climate-smart agriculture, breadfruit is a candidate for carbon market projects. In general, it is mandated that agricultural, aquacultural, agro-forestry, and forestry projects must satisfy a carbon registry's approved carbon accounting protocols. These protocols derive from science-based publications of governmental and non-governmental organizations such as the International Panel on Climate Change (IPCC) and Verified Carbon Standard (VCS). However, carbon-accounting protocols and the development of carbon market projects remain in their infancy, and protocols and methodologies do not yet exist to cover all situations. In particular, tropical and indigenous crops, including breadfruit, are understudied and poorly represented in existing protocols and models [11,12]. Therefore, although carbon project methodologies have been developed, tailoring existing protocols to specific situations, such as breadfruit agroforestry, still requires additional research.

Accurate assessment of estimated forest biomass is vital for applications such as wood extraction, tracking changes in forest carbon stocks, and the global carbon cycle. Allometric models—mathematical functions that connect the dry mass of a tree with one or more tree dimensions, such as height, diameter, and wood density—are still the dominant method for assessing tree biomass, and building diverse databases is important for improving accuracy and application [13–15]. Allometric equations tend to have good prediction performance represented by high $r^2$ values and are traditionally derived by destructive sampling of entire trees, which, although considered highly accurate, is expensive and time-consuming [14,16,17]. Other methods include obtaining the tree volume through detailed measurements or remote sensing technologies such as LiDAR or photogrammetry [18,19]. Statistical methods tend to apply scatter plots of individual trees to explore data trends, with emphasis on the correlation coefficients and associated errors [20,21].

This paper applies carbon accounting methodologies to develop quantitative data on the terrestrial carbon pools of breadfruit in Hawai'i, which, as an ecologically diverse subtropical site, offers opportunities for extrapolation to other regions of the world. In order to assess the carbon sequestration potential of breadfruit, we pursued the following objectives: (1) to develop an allometric relationship of breadfruit wood/carbon density, (2) to develop an allometric equation of breadfruit tree volume, (3) to develop an allometric equation of breadfruit foliar biomass, (4) to characterize the growth curves for breadfruit, and (5) to describe above-ground carbon sequestration in breadfruit through landscape-level extrapolations. This effort will inform future work and discussions of a potential role for breadfruit in climate-friendly agriculture, including the development of breadfruit-focused carbon projects, and provide tools for the quantification of these carbon pools.

## 2. Materials and Methods

### 2.1. Overview of Sampling Approaches

To develop an appropriate allometric equation for *A. altilis* in Hawai'i, this project collected detailed architectural measurements to estimate the total volume of unadulterated breadfruit trees, wood, and leaf samples from trunks and stems of various diameters to calculate wood density and carbon density, and leaf biomass to relate it to terminal branch size [14,16,22]. Between August 2020 and March 2023, breadfruit trees of the cultivar 'Maoli' were opportunistically sampled on Hawai'i Island in the districts of Hilo, Hāmākua, and Kona. All sampling occurred on private lands and specific locations are not disclosed to protect the landowners. Detailed architectural data were collected following the procedures used in other minimally destructive study methods, in which diameter measurements were

collected every 1 m along all woody material of the entire tree. A mathematical model was created in Excel to calculate the volume of each segment and the total tree volume. Wood samples were collected from branches of varying diameters from four trees and were used to analyze wood density and carbon density, which were subsequently used to generate regressions against branch diameter. These relationships were integrated with the tree volume model for each segment based on the average stem diameter to calculate total stem biomass and carbon. Foliar mass and carbon were obtained by collecting terminal branches of various sizes from eight trees. Leaf biomass was determined for all leaves on the branch and regressed against the terminal branch base diameter. This relationship was integrated into the mathematical model to calculate leaf biomass and carbon based on each terminal branch diameter. The sum of all components generated total above-ground biomass (AGB) and carbon for each tree. The total AGB for each tree was regressed against the diameter at breast height (DBH) to determine an allometric relationship described by a non-linear equation. Separately, the DBH of 208 breadfruit trees of known age was measured. Trees were classified by habitat suitability and DBH regressed against age to determine growth equations. Growth equations were combined with the allometry to make landscape-level extrapolations about potential carbon sequestration in breadfruit orchards. Overall, this study relies on relatively small sample sizes for tree biomass ($n = 12$), wood and carbon density ($n = 28$), leaf biomass ($n = 40$), and growth curve ($n = 208$). This study initially sought to identify the best published allometric equation to apply to breadfruit. Despite the low sample size, robust allometric correlations were observed.

### 2.2. Woody Biomass Volumetric Measurements

The major and minor diameter was measured and recorded every meter for the main trunk and all branches of each tree, starting at the ground surface. For terminal branch measurements, the length of the final segment was recorded along with the branch diameter 5 cm from the terminus. The diameters were measured using tree calipers measured to the nearest mm, and the length of each segment was measured with a cloth measuring tape to the nearest cm. For each branch, the segment number was recorded as the sequential measurement, so that the series of measurements for each branch proceeded numerically (1, 2, 3...) and the count reset for each new branch. These measurements (major and minor diameter, distance along the branch from the previous measurement, and segment number) were used to calculate the total tree volume using a spreadsheet model described below.

### 2.3. Wood Density and Carbon Content

To determine wood and carbon density, "cookies", or cross-sections of branches of various diameters, were opportunistically collected where destructive sampling was possible. The volume ($cm^3$) of the wood cookies was determined using the water displacement method, and the samples were then oven-dried until constant mass and weighed (g). Wood density ($g/cm^3$) for each cookie was calculated by dividing dry mass by volume. To determine carbon density, holes were drilled through the cross-section of each cookie. The wood shavings were collected, pulverized, and encapsulated for total carbon concentration (C) analysis using an Elemental Analyzer. Carbon density ($gC/cm^3$) was determined by multiplying the wood density by %C. Wood and carbon density were each regressed against the average branch diameter to describe wood density and carbon as a function of branch diameter.

### 2.4. Foliar Biomass Estimation

Forty branches of varying diameters were harvested from trees, severing them at the branch collar. The base and terminal diameter of the branch were measured. All leaves were removed and oven-dried, then weighed to determine the total dry foliar biomass from each branch. Total dry foliar biomass was regressed against the base and the terminal branch diameter. Leaves from a subset of branches were pulverized, encapsulated, and analyzed for total carbon concentration (%C) using an Elemental Analyzer.

*2.5. Mathematical Model of Tree Biomass/Carbon*

Linear and non-linear regressions were used to examine the relationship between parameters of interest, as described in Sections 2.2–2.4. Analyses were conducted using JMP (SAS Institute; Cary, NC, USA), with the coefficient of determination ($r^2$) and probability values ($P$) used to describe the accuracy of the mathematical equations. A spreadsheet model was used to combine all factors measured and computer total biomass and carbon for each tree measured. For each row in the spreadsheet, the major ($D_M$) and minor ($D_m$) diameters were used to calculate the average radius ($R$) in cm and the cross-sectional area ($A$) in cm$^2$, as shown in Equations (1) and (2).

$$R = \frac{D_m + D_M}{4} \tag{1}$$

$$A = \pi \times \frac{D_M}{2} \times \frac{D_m}{2} \tag{2}$$

Leveraging the branch segment number ($S$), an *IF* statement was used to calculate the volume of each segment in the tree, as described in Equation (3).

$$V_n = IF[S_n > S_{n-1}], \ \left( \frac{A_n + A_{n-1}}{2} \times L_n \right), \ 0 \tag{3}$$

where $V$ is the volume in cm$^3$. For each segment, the regression equations we determined to describe the relationship of wood density ($\rho$) and carbon concentration ($C_\%$) as a function of branch radius were applied, so that total carbon density was considered a continuous function of branch thickness, as in Equations (4)–(6).

$$C_w = V * \rho * C_\% \tag{4}$$

$$\rho = 0.1898 + 0.00986 \times \left( \frac{R_n + R_{n-1}}{2} \right) \tag{5}$$

$$C_\% = 42.81 + 0.3157 \times \left( \frac{R_n + R_{n-1}}{2} \right) \tag{6}$$

where $C_w$ is the total carbon mass of each woody component in g. For each branch, total leaf biomass was calculated by applying the regression equation we determined to describe the relationship between leaf biomass ($M_l$) and branch base diameter ($D_b$), with total carbon calculated as the product of leaf biomass and the average leaf carbon density. We again leveraged branch segment number to apply an IF statement, so that the model only calculates leaf biomass for the first (base) measurement of each branch as in Equation (7):

$$M_l = IF[S_n < S_{n-1}], \ [0.514 \times (-102.1 + 103.5 \times R_n)], \ 0 \tag{7}$$

$$C_l = M_l \times 0.514 \tag{8}$$

where $M_l$ is the total dry biomass of the leaf component in g, and $C_l$ is the total carbon mass of leafy components in g.

Woody and/or leaf biomass was calculated in Excel for each row of the spreadsheet. Total tree biomass and total tree carbon were calculated as the sum of all individual woody segments and leaf components for all rows of each tree. Total tree biomass/carbon was compared against tree DBH, and a non-linear regression was used to characterize the relationship between tree DBH and total tree biomass/carbon. For all relationships, the $r^2$, $P$, and root mean square errors were assessed.

### 2.6. Breadfruit Growth Rates

Developing equations to describe changes to tree diameter at breast height (DBH) as the tree grows is fundamental to tree growth models; calculating the change in DBH over time is needed to estimate biomass growth over time and consequently carbon sequestration and storage [23]. A wide range of environmental parameters are influential to plant growth including temperature, soil type and fertility, rainfall and water availability, solar radiation, and many others [24]. Typically, the most limiting factor is considered to define the rate of tree growth, and habitat modeling for breadfruit and other trees tends to take the approach of understanding the most limiting environmental parameter (e.g., [25]). However, different factors may be limiting across different time periods, both within annual cycles and across inter-annual variation (e.g., [26]). Typical protocols for determining tree growth rates involve measuring the desired tree parameters periodically over time. However, space-for-time substitution may be used if trees of multiple ages are accessible to be measured. We measured the DBH of 208 *A. altilis* trees of known age from five regions on the Big Island of Hawai'i. Location was recorded and habitat suitability was determined in ArcGIS by extracting the locational data from the habitat suitability map generated by [25]. Trees were classified into categories of suitability as "High" (>81), "Medium" (70–81), and "Low" (<70) for analysis.

### 2.7. Below Ground Biomass (BGB) and Landscape-Scale Total Biomass over Time

To extrapolate the potential for breadfruit carbon sequestration over time, we employed the best-fit growth relationship for the "high" habitat suitability, assuming areas of high suitability would be preferentially developed, to determine projected DBH at 5, 10, 15, and 20 years. Projected DBH values were inputted into the allometric equations we determined to estimate total tree AGB and carbon mass. Projected AGB values were combined with a published root-to-shoot ratio and generic carbon density to estimate the below-ground biomass (BGB) and associated carbon. Total carbon storage by a breadfruit tree was calculated as the sum of the above-ground and below-ground components. To move to the landscape scale, we used industry guidelines of a planting density of 120 trees per hectare to create an estimate of total carbon stocks and convert to associated carbon dioxide sequestration per hectare at the projected time points by multiplying total carbon by multiplying by 3.67. Error propagation was conducted by taking the square root of the sum of squares of all root mean square errors associated with the regression equations applied, and converting where necessary, to provide an estimate of error for the landscape-scale projections of carbon sequestration and storage associated with breadfruit cultivation.

## 3. Results

### 3.1. Wood Density and Carbon

Cross-section "cookies" from 28 branch samples ranging in average radius from 0.7375 cm to 16.625 cm were harvested and measured from four 'Maoli' breadfruit trees. Wood density exhibited a generally linear increase with increasing radius of the branch/trunk (Figure 1). The relationship between wood density as a function of branch radius is fairly robust ($r^2 = 0.799$). The relationship of wood density to average radius was used as the function described in Equation (5).

Carbon concentration was measured for the branch cross-sections. The mean for all samples was 44.8% C. Percent carbon as a function of radius demonstrates a slight increase with branch diameter (Figure 2a). Although the relationship appears to have some non-linear behavior, the application of various non-linear regressions only marginally improved the $r^2$ values. Therefore, the linear regression was applied to describe carbon concentration as a function of stem diameter, as described in Equation (6). Despite the variation in carbon concentration, a calculation of carbon density ($gC/cm^3$) by average branch radius yields a highly significant relationship with a robust ($r^2 = 0.865$).

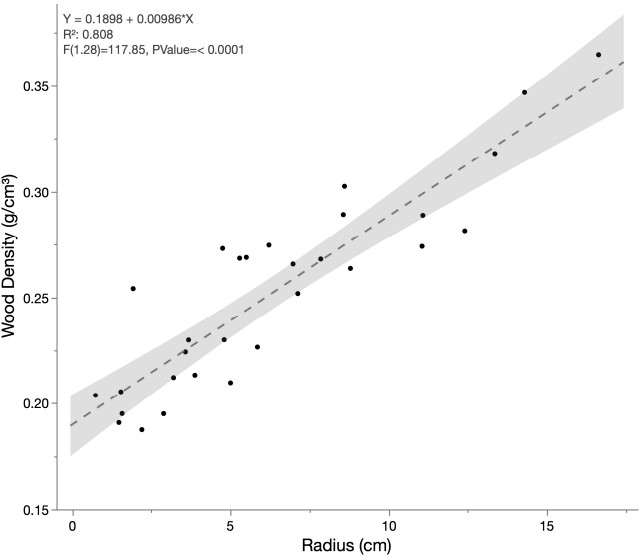

**Figure 1.** Dry wood density as a function of average branch/trunk radius.

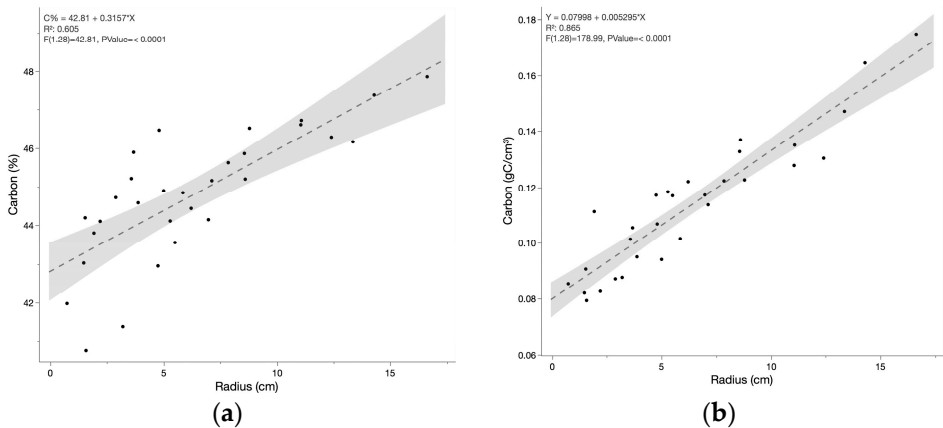

**Figure 2.** Carbon concentration in dried breadfruit wood samples as a function of average branch radius in (**a**) percent carbon by mass and (**b**) carbon mass by volume.

### 3.2. Foliar Biomass

Foliar biomass per terminal branch was described as a function of stem diameter. The foliar biomass from 38 terminal branches was collected, dried, and weighed for total biomass. The base and tip diameters (5 cm from the branch terminus) of the terminal branches were recorded alongside these weights. The total number of leaves per terminal branch ranged from 4 to 14 with total dry weight ranging from 3.96 g to 262.85 g. The relationship between foliar dry-weight biomass and the stem diameter is shown in Figure 3 below. Dry weight biomass was more strongly correlated with the stem base diameter (Figure 3; $r^2 = 0.907$) than the tip diameter ($r^2 = 0.819$). The associated regression equation was used to describe leaf biomass as a function of terminal branch base diameter in Equation (7). Carbon percentage for the foliar biomass was determined for half ($n = 20$) of the samples and was found to have an average of 51.4% with a standard deviation of 1.14.

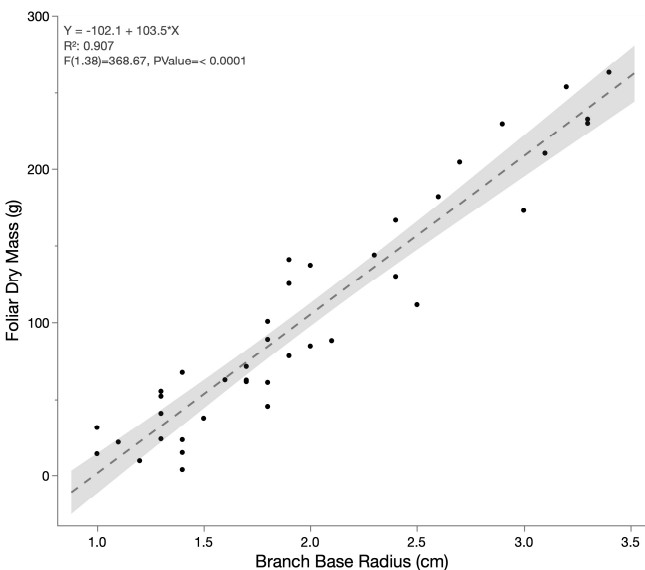

**Figure 3.** Foliar biomass of terminal stems as a function of the stem base diameter for 38 branches.

### 3.3. Woody Biomass Volume and Total Above-Ground Biomass (ABG)

Using the method described, measurements from 12 *A. altilis* cv. 'Maoli' trees ranging in DBH from 2.6 cm to 41.2 cm were collected from Hawai'i Island. Total volume, as calculated as a sum of the branch segments, ranged from 1042 cm$^3$ to 973,860 cm$^3$. The resulting woody biomass from our spreadsheet model ranged from 216 g to 350,608 g, and the total foliar biomass ranged from 177 g to 36,395 g. The sum of woody and foliar biomass was used to represent the total ABG (Table 1). The total ABG for each tree was regressed against the measured DBH of those trees, demonstrating a robust non-linear relationship best described by a quadratic equation (AGB = $-4.586 + 0.1635 \times$ DBH $+ 0.2229 \times$ DBH$^2$; r$^2$ 0.98, $p < 0.001$).

**Table 1.** Calculated dry wood biomass, leaf biomass, and total above ground biomass (ABG) of breadfruit trees measured in this study.

| DBH (cm) | Wood Biomass (g) | Leaf Biomass (g) | Total AGB (kg) |
| --- | --- | --- | --- |
| 2.6 | 216 | 177 | 0.39 |
| 4.7 | 1575 | 324 | 1.90 |
| 6.2 | 4446 | 854 | 5.30 |
| 7.6 | 5520 | 1287 | 6.81 |
| 10.9 | 11,971 | 3851 | 15.82 |
| 14.4 | 32,452 | 10,054 | 42.51 |
| 17.1 | 59,815 | 14,139 | 73.95 |
| 22.4 | 83,629 | 23,186 | 106.81 |
| 26.9 | 124,470 | 30,854 | 155.32 |
| 27.4 | 144,833 | 33,566 | 178.40 |
| 35 | 232,102 | 36,394 | 268.50 |
| 41.2 | 350,608 | 31,811 | 382.42 |

### 3.4. Growth Curves for Breadfruit

Of the trees surveyed, 95 were the 'Maoli' and 113 were the 'Maafala' variety. Across all trees, the linear regression of diameter against age was indistinguishable between the two varieties, and therefore subsequent analysis combined the two varieties. Across all trees, the relationship between growth and age was best described by a linear function (DBH = $5.867 + 1.013 \times$ Age, r$^2$ = 0.87, $p < 0.001$). However, this represents highly diverse habitats across Hawai'i Island. Trees were therefore broken into suitability classes of High

(n = 38), Moderate (n = 42), and Low (n = 128) [25]. Regressions by suitability classification were best explained by quadratic functions (Figure 4; Table 2).

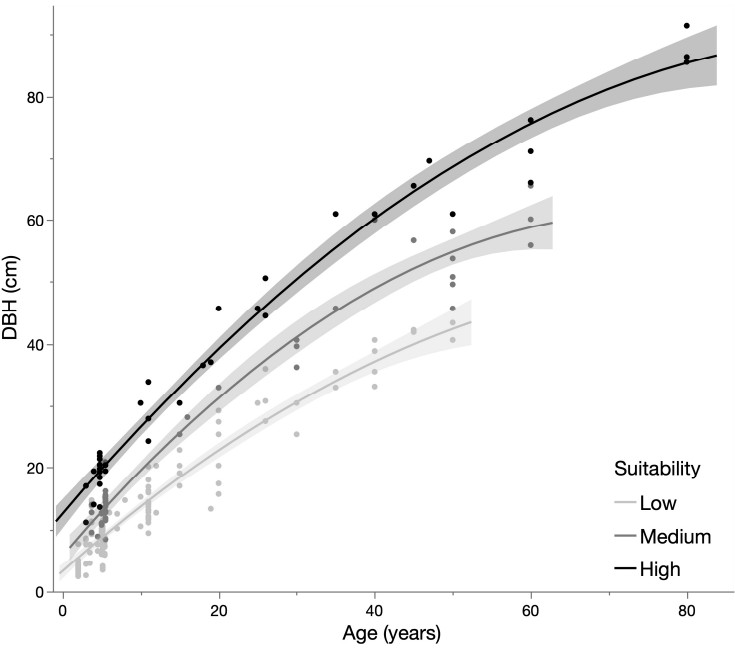

**Figure 4.** Growth curves by suitability classification for breadfruit.

**Table 2.** Equations describing the growth in DBH of 208 breadfruit trees on Hawai'i Island classified by habitat suitability.

| Suitability Class | Equation | $R^2$ | RMS Error |
|---|---|---|---|
| High | DBH = 12.57 + 1.47(Age) − 0.007(Age)$^2$ | 0.98 | 3.8 |
| Medium | DBH = 5.766 + 1.47(Age) − 0.010(Age)$^2$ | 0.96 | 3.6 |
| Low | DBH = 3.308 + 1.07(Age) − 0.0056(Age)$^2$ | 0.89 | 3.3 |

### *3.5. Landscape Extrapolations*

To predict breadfruit growth over time, only the high suitability class was used for extrapolation, as noted in Equation (9), assuming that areas of high suitability would be preferentially developed for the breadfruit industry.

$$DBH = 12.57 + 1.47 \times Age - 0.007 \times Age^2 \tag{9}$$

The growth formula was used to estimate the diameter at breast height (DBH) of trees at ages 5, 10, 15, and 20 years. The resulting DBH values were applied to the allometric equation to estimate the total above-ground biomass (AGB), as described in Equation (10).

$$AGB = -4.586 + 0.1635 \times DBH + 0.2229 \times DBH^2 \tag{10}$$

A root-to-shoot ratio was applied to the resulting AGB value generated using a generic conversion for tropical/subtropical moist forest/plantation environments (0.235) as published by [27] to estimate total below-ground biomass (BGB). For above-ground carbon we applied our carbon-specific allometric equation, and for below-ground carbon we applied a generic carbon ratio of 0.4 for convert biomass to carbon. The sum generates an estimate of the total carbon stock of a single tree. To extrapolate to a per hectare basis we applied the industry standard of 120 trees per hectare. The results from these extrapolations are summarized in Table 3.

**Table 3.** Extrapolation of total $CO_2$ sequestration associated wtih a breadfruit orchard based on the growth curve, allometric equation, and wood/carbon density determine in this study.

| Age | DBH (cm) | AGB (kg) | BGB (kg) | AGC (kg) | BGC (kg) | C (kg/Tree) | $CO_2$ (tons/ha) |
|---|---|---|---|---|---|---|---|
| 5 | 19.7 | 85.5 | 20.5 | 39.2 | 8.2 | 47.4 | 16.7 |
| 10 | 26.6 | 157.1 | 37.7 | 73.3 | 15.1 | 88.3 | 31.2 |
| 15 | 33.0 | 244.2 | 58.6 | 115.0 | 23.4 | 138.4 | 48.8 |
| 20 | 39.2 | 343.8 | 82.5 | 162.9 | 33.0 | 195.9 | 69.1 |

## 4. Discussion

The allometric equation developed through this study is comparable to previously published allometry (Figure 5) [17,28,29]. The results from this study align very well with the only other equation published for an *Artocarpus* spp. (jackfruit; *Artocarpus heterophyllus*) [29]. The number of trees suggested for developing an allometric equation ranges from 17 to over 100 [13,30]. Although both the previous study on jackfruit and our current study represent relatively small sample sizes, the alignment of the studies suggests that Artocarpus AGB falls within this range. These ABG values determined for Artocarpus are considerably lower than the generic pan-tropical equations for broadleaf species for either wet or moist conditions (Figure 5). We suggest that this makes sense, given that breadfruit is well known as a "light" wood, being used in traditional Polynesian culture for its lightness and ability to float and therefore being used to make fishing floats, the gunnels on canoes, and surfboards because of its characteristics [11]. The "lightness" of the wood corresponds to a lower wood density and, accordingly, a lower carbon density on a per-volume basis compared to many tropical woods. The relatively lower carbon sequestration and storage of breadfruit is something to be considered in its application as a climate-smart commodity, although the ABG sequestration potential of breadfruit as a perennial tree is still considerably higher than comparable annual staple starches [31,32].

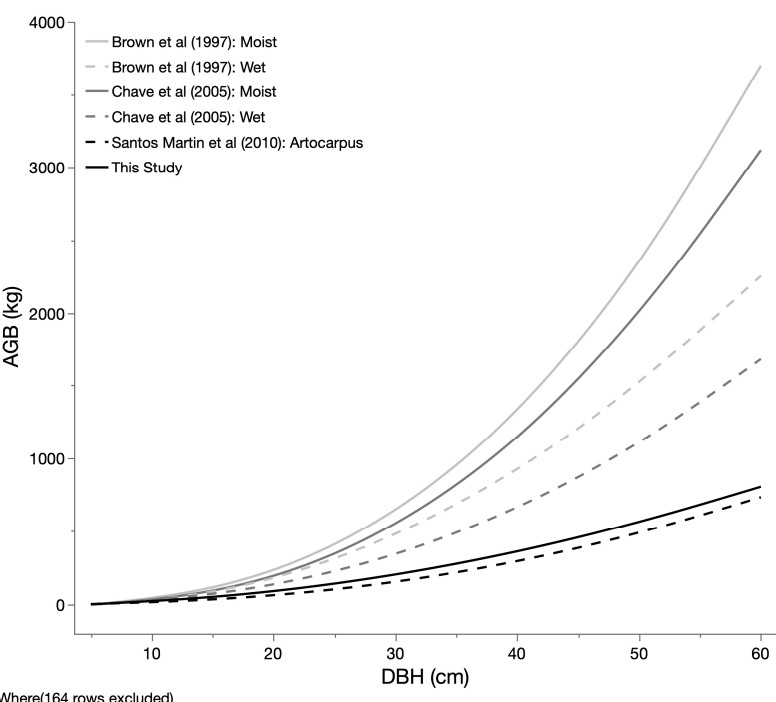

Where(164 rows excluded)

**Figure 5.** A comparison of published allometric equations that are relevant to breadfruit biomass accumulation [17,28,29].

Even within a single species, site-specific factors can cause variations in tree parameters that cause significant disruption to the fit of the model to specific sites [33]. This study

demonstrates that minimally destructive methods can enable the development of allometric equations that align closely with destructive methods. This study specifically selected trees that were (1) grown in areas of high productivity, (2) were generally "alone", receiveing full sun and with adequate space to grow without effects from neighboring trees, and (3) were unadulterated in that there were no pruning, major loss of limbs over time, or other major impediments on their growth. While the trees were selected in order to support a clean allometric relationship, these trees selected represent relatively optimal growing conditions and therefore represent the higher end of carbon sequestration. This is unlikely to represent true orchard conditions, where trees would be pruned to manage the tree height to facilitate harvest.

Applying wood and carbon density as a continuous function of branch diameter may improve estimates of total carbon, and should be explored further as a standardized method for carbon accounting. In this study, the incorporation of a continuous relationship was afforded because of the non-destructive methods employed, which required the tedious measurements of all branch diameters. However, as imaging methods, such as photogrammetry and LiDAR, become increasingly improved and are applied more frequently, the opportunity to carry out mathematical integration will also increase. Therefore, non-destructive methods may also afford new opportunities to improve the accuracy of total carbon calculations.

Strong relationships between age and DBH were shown using tree age and previously defined habitat suitability, suggesting that good predictions about growth rates across environments can be determined for breadfruit. It is important to note that tree samples for both the growth rates and the determination of AGB were unadulterated trees. This is unlikely for a plantation scenario, in which trees are generally pruned and otherwise actively managed. This could both increase (e.g., through fertilization) or decrease (e.g., through pruning) growth rates, and would likely decrease AGB (through pruning). These are activities that need to be considered in future work. However, this initial analysis demonstrates the potential to provide an accurate assessment and prediction of breadfruit growth and carbon storage.

The growth rates were utilized in conjunction with our previously determined allometry, published root-to-shoot ratio, and industry standards to demonstrate a landscape-level prediction of carbon sequestration in the biomass associated with breadfruit plantations. This shows that breadfruit can sequester substantial amounts of carbon in solely the most direct and measurable pool of biomass. Although the sequestration and storage of carbon in breadfruit trees are substantially higher in breadfruit trees than in comparable staple crops, a substantial scale would still be required before reasonable returns might be realized from the carbon outcomes. For instance, if we assume a set cost of $2000 for paying a third party carbon verifier for a baseline assessment and monitoring every five years over 20 years ($10,000), and a current middle-of-the-road carbon price of $5 per metric ton, then a 26.8 ha plantation would be required just to break even for the carbon payments to cover the cost of monitoring over a twenty year period. For a 100 ha orchard, this would only equate to $35,000 in revenues and $25,000 in net revenues. In comparison, if we assume a modest 136 kg yield per tree [11] and a current farmgate price of $2.75 per kg, this would equate to ~$4,488,000 in fruit per year at a wholesale price, or $67,000,000 over the same time (assuming a five-year ramp-up period without fruiting). This suggests that over a 20-year project, carbon would account for 0.05% of the revenues related to fruit. Such extrapolations suggest that the price of carbon in existing markets are likely not substantial enough to influence crop decisions.

While our extrapolations are based on monocropped orchards of breadfruit, we emphasize that breadfruit is often grown in diversified, co-cropped settings and encourage diversified agroforestry methods for the ecological and social benefits that they can support. In traditional Polynesian cropping systems, a broad range of agroforestry methods were employed, many of which relied on breadfruit as a core component [3,34]. Contemporarily, breadfruit is often grown in agroforestry settings, supporting a broad range of ecological

and social outcomes [35–37]. Among the enhanced benefits of diversified agroforestry include additional carbon sequestration in both biomass and soil carbon stocks [38]. While the study of breadfruit allometry provides a starting point for understanding carbon in such systems, additional work is essential to support agricultural transitions to more sustainable systems of food production.

Future work could better describe how carbon percentage varies over time and by variety, and in particular pruned trees should be studied to better represent orchard management. This would make the carbon estimate more robust and could point to ways of better designing carbon projects for agriculture.

**Author Contributions:** Conceptualization, C.L. and N.K.L.; methodology, C.L.; software, N.K.L.; formal analysis, C.L.; investigation, N.K.L. and C.L.; resources, N.K.L.; data curation, N.K.L.; writing—original draft preparation, C.L.; writing—review and editing, N.K.L.; visualization, N.K.L.; supervision, N.K.L.; project administration, N.K.L.; funding acquisition, N.K.L. and C.L. All authors have read and agreed to the published version of the manuscript.

**Funding:** This research was funded in part by the USDA National Institute of Food and Agriculture, McIntire Stennis Project (8050-MS) and the National Science Foundation CAREER grant #1941595.

**Institutional Review Board Statement:** Not applicable.

**Informed Consent Statement:** Not applicable.

**Data Availability Statement:** All data is available by request from the corresponding author.

**Acknowledgments:** We thank Dolly Autufuga, Thomas Haensel, Alex Curtis and the members of the Indigenous Cropping Systems Laboratory for helping with the field work.

**Conflicts of Interest:** The authors declare no conflict of interest.

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
