# Peer review of "Determining Allometry and Carbon Sequestration Potential of Breadfruit (Artocarpus altilis) as a Climate-Smart Staple in Hawai‘i"

_sustainability, doi:10.3390/su152215682_

Round 1
Reviewer 1 Report
Comments and Suggestions for Authors
The manuscript presented an interesting study related to the development of
Allometry equation of breadfruit. However, the introduction is very poor, and the authors didn’t present the state of art and the limitation, and the approach used in the development of allometry equation for forestry project. The discussion needs to be improved.
General comments:
· The state of art need to be developed by giving more details about the models and approach used to carbon sequestration potential agroforestry
· Resolution of all figures need to be improved.
· Section “Conclusion” need to be added.
· Photos and illustration need to be added in the methodology.
· Add the map of study region.
Line 15: remove the word “show”.
Line 16: replace “suggests” by “shows”
Line 24: don’t use abbreviation for the first time “GHG”
Line 77: change the title “Overview” is not very clear you can replace it by “Sampling and Biomass measurement”.
Line 111: add more details about the precision of the calipers used.
Line 112 : remove this sentence , it is not relevant ‘The measurements were input into a Microsoft Excel spreadsheet….”
Line 131 : Remove these sentence “Regression analysis was used to explore the relationship between foliar biomass and terminal branch diameter.” The same comments in line 137.
Add a section of the Statistical analysis to give all details about the statistics tests performed during this study.
Line 224 : improve the resolution of the Figure 1.
Line 215 : correct “0.7375”
Line 217 : don’t use abbreviation in the first time “ r2 ”
Line 297 and 303 : the equations need to be presented in the methodology
Comments on the Quality of English Language
Moderate language correction is needed.
Author Response
- The state of art need to be developed by giving more details about the models and approach used to carbon sequestration potential agroforestry
We added the following paragraph to the Introduction:
Accurate assessment of estimated forest biomass is vital for applications such as wood extraction, tracking changes in forest carbon stocks, and the global carbon cycle. Allometric models—mathematical functions that connect the dry mass of a tree with one or more tree dimensions, such as height, diameter, and wood density—are still the dominant method for assessing tree biomass and building diverse databases is important for improving accuracy and application [15]–[17]. Allometric equations tend to have good prediction performance represented by high r2 values, and are traditionally performed by destructive sampling of entire trees, which, although considered very accurate, is expensive and time-consuming [16], [18], [19]. Other methods include obtaining the tree volume through detailed measurements or remote sensing such as LiDAR or photogrammetry [20], [21]. Statistical methods tend to apply scatter plots of individual trees to explore data trends, with emphasis on the correlation coefficients and associated error [22], [23].
- Resolution of all figures need to be improved.
The resolution of all figures was increased to 300 dpi.
- Section “Conclusion” need to be added.
Per the template provided, the conclusion is optional, (This section is not mandatory but can be added to the manuscript if the discussion is unusually long or complex). We feel that this was a straightforward study and our thoughts were already summarized within the discussion, and therefore did not necessitate a conclusion.
- Photos and illustration need to be added in the methodology.
We do not have photos and illustrations of the methodology, but there are already illustrations published in the resources that we cite in the methods.
- Add the map of study region.
We do not see much value in this, and since the individual trees were secured on private lands we do not wish to show the precise locations of the trees.
Line 15: remove the word “show”.
Done
Line 16: replace “suggests” by “shows”
Done
Line 24: don’t use abbreviation for the first time “GHG”
Done
Line 77: change the title “Overview” is not very clear you can replace it by “Sampling and Biomass measurement”.
Changed to Overview of Sampling Approaches
Line 111: add more details about the precision of the calipers used.
Added “to the nearest mm”
Line 112 : remove this sentence , it is not relevant ‘The measurements were input into a Microsoft Excel spreadsheet….”
Done
Line 131 : Remove these sentence “Regression analysis was used to explore the relationship between foliar biomass and terminal branch diameter.” The same comments in line 137.
Done
Add a section of the Statistical analysis to give all details about the statistics tests performed during this study.
Added: “For all relationships, the r2, p, and root mean square errors were assessed.”
Line 224 : improve the resolution of the Figure 1.
Done
Line 215 : correct “0.7375”
Done
Line 217 : don’t use abbreviation in the first time “ r2 ”
This is a ridiculous suggestion…I have never even seen “R squared” or “the proportion of variance for a dependent variable" written out in a scientific paper.
Line 297 and 303 : the equations need to be presented in the methodology
They are not preexisting equations. They cannot be presented in the methodology, because they are derived from the results….we could not show where the equations came from without first presenting the findings.
Reviewer 2 Report
Comments and Suggestions for Authors
Manuscript entitled Breadfruit (Artocarpus altilis) as a climate-smart staple: Allometry and carbon sequestration potential is well written and the structure of the manuscript is good. But there are some concerns which need to be addressed before acceptance
Title is too general, looks like a general review article. Modify it please
Abstract is too short, it should be of at least 150 words
Add scientific name of breadfruit in abstract
At the end of the introduction section, the hypotheses and objectives are not clearly stated.
Need to revisit the abbreviation, all the abbreviation should be given full form when they appear first time in the manuscript
Discussion should be merely based on the observed findings. It should not only be the literature review. Answer the question posed in introduction and correlate your finding with the existing knowledge.
The conclusion is the right place to highlight novel results; in its present form, none have been presented and it should be supported by future outcomes in lieu of the current research.
Comments on the Quality of English LanguageLanguage editing is required, there may be spell and grammatical mistakes creeping within the text, authors must check and correct them.
Author Response
Title is too general, looks like a general review article. Modify it please
Title was edited to read “Determining allometry and carbon sequestration potential of breadfruit (Artocarpus altilis) as a climate-smart staple in Hawai’i.”
Abstract is too short, it should be of at least 150 words
We have added details of the results to the abstract.
Add scientific name of breadfruit in abstract
Done
At the end of the introduction section, the hypotheses and objectives are not clearly stated.
We edited the last paragraph of the introduction to clearly state the objectives. This is inductive research, in which the data is meant to explore an area rather than prescribe to a specific hypothesis.
Need to revisit the abbreviation, all the abbreviation should be given full form when they appear first time in the manuscript
Done
Discussion should be merely based on the observed findings. It should not only be the literature review. Answer the question posed in introduction and correlate your finding with the existing knowledge.
We discussed the findings in light of previous studies, and even have a figure that specifically compared our results to previous findings. We answered the objectives sought out in the introduction, which is to evaluate breadfruit, listed as a priority crop for climate-smart agriculture, in terms of carbon sequestration potential. The review will have to be more specific in order to address this comment.
The conclusion is the right place to highlight novel results; in its present form, none have been presented and it should be supported by future outcomes in lieu of the current research.
Per the template provided, the conclusion is optional, (This section is not mandatory but can be added to the manuscript if the discussion is unusually long or complex). We feel that this was a straightforward study and our thoughts were already summarized within the discussion, and therefore did not necessitate a conclusion.
Reviewer 3 Report
Comments and Suggestions for Authors
1. Revise the abstract. Please add main finding with some significant fact and figures. In the current for it is over generalized and should be improved.
2. The author should also include novelty statement in the introduction section, and how your work is different from previously conducted research on more or less same topic.
3. Heading 2.2 to 2.5 , no reference was sighted. same goes with the heading 2.7.
I will urge the authors to cite proper reference in the methodology section.
Moreover, the author should justify the sample size used in this study. I think it is too small and field data should be of more then one year.
4. Discussion section also seems revolving around author's own opinion. However, a critical comparison is missing. Moreover, the information provided in the discussion in many cases is not supported by any suitable reference for example from line 351-393.
I will suggest thorough revision of the work.
5. Please provide a valid conclusion of the work. Conclusion section is missing.
Author Response
- Revise the abstract. Please add main finding with some significant fact and figures. In the current for it is over generalized and should be improved.
Main finding were added to the abstract.
- The author should also include novelty statement in the introduction section, and how your work is different from previously conducted research on more or less same topic.
The introduction states that breadfruit has been identified as a priority underutilized species for climate mitigation, and yet, as with most indigenous crops, are understudied, including its allometry or carbon sequestration potential. This is the novelty of the study.
- Heading 2.2 to 2.5 , no reference was sighted. same goes with the heading 2.7. I will urge the authors to cite proper reference in the methodology section.
The citations in Section 2.1 are used to support the relevant methods used in 2.2-2.5. We describe our methods in detail.
Moreover, the author should justify the sample size used in this study. I think it is too small and field data should be of more then one year.
Field data was collected over several years. We added a description of the sample size and justification: "Overall, this study relies on relatively small sample sizes for tree biomass (n=12), wood and carbon density (n=30), leaf biomass (n=40), and growth curve (n=208). This study initially sought to identify the best published allometric equation to apply to breadfruit. Despite the low sample size, robust allometric correlations were observed."
- Discussion section also seems revolving around author's own opinion. However, a critical comparison is missing. Moreover, the information provided in the discussion in many cases is not supported by any suitable reference for example from line 351-393.
Results are directly compared to previous studies.
- Please provide a valid conclusion of the work. Conclusion section is missing.
Per the template for the journal, “This section is not mandatory but can be added to the manuscript if the discussion is unusually long or complex.” We felt that this study is quite straightforward and does not require a conclusion.
Round 2
Reviewer 1 Report
Comments and Suggestions for Authors
Almost all of the comments presented in the first round were not addressed by the authors (methodology and conclusion are still missing).
The authors need to understand that the methodology needs to be documented and illustrated.
The document needs to be revised by senior scientists before submission.
Comments on the Quality of English Language
English correction is needed.
Author Response
Aloha e -
We unfortunately find this review extremely unhelpful and impossible to respond to. The Reviewer indicates that "almost all of the comments presented in the first round were not addressed by the authors (methodology and conclusion are still missing)." Yet we provided a point by point response to every issue that the Reviewer raised in the first round of reviews. The Reviewer provides minimal indications as to which points were not addressed. As we already mentioned, per the journal template the conclusion is not required except for exceptionally complex papers/discussions, which we believe our paper is decidedly not. The Reviewer unfortunately provides no explanation why they believe a conclusion is needed, nor and guidance suggestion as to what should be in the conclusion. The suggestion that a conclusion should exist just for the sake of it is not a compelling reason to the authors. As the the methodology, the first review only indicated that "Photos and illustrations need to be added in the methodology." We argued that we are not illustrators and that we have adequately described in writing the methods. Furthermore, we have reviewed over two dozen recent papers on allometry, not one of which provides illustrations for the methodology, and only half of which provides maps of the sites. As already indicated in our paper, our sites were on private lands and we will not provide a detailed map of the trees in order to protect the the location of the trees. Although in the first Review the only request on the methodology was in regards to illustrations, the Reviewer in now indicating that the methodology itself is "missing" in your first comments, and "needs to be documented" in your second comment. As the methodology is most decidedly present, well-documented in review by colleagues, and was adequate in the first round of reviews, we stand by our decision that there is nothing more that needs to be done in response.
The Reviewers other comment is only that "The document needs to be revised by senior scientists before submission." This is not any sort of constructive review, and furthermore is insulting. The paper is authored by a senior scientist with over 50 peer-reviewed publications, two books, two edited book, and four edited journal volumes including one in this journal (Sustainability). This comment is in no way constructive and is clearly meant to belittle the authors.
Finally, although providing no examples or evidence, the Reviewer indicates that the paper requires "extensive" editing of the English language. We have given the paper to an English professor who found "a couple of typos, but no egregious English errors."
Reviewer 3 Report
Comments and Suggestions for Authors
comments were addressed by the authors.
Comments on the Quality of English LanguageSatisfactory
Author Response
Thank you for your review of the manuscript. Your insightful comments were very helpful in improving the document.